# A Universal Semantic-Geometric Representation for Robotic Manipulation

**Tong Zhang**[1,2,3*]    **Yingdong Hu**[1,2,3*]    **Hanchen Cui**[3]    **Hang Zhao**[1,2,3]    **Yang Gao**[1,2,3†]

[1]Tsinghua University    [2]Shanghai Artificial Intelligence Laboratory    [3]Shanghai Qi Zhi Institute
{zhangton20,huyd21}@mails.tsinghua.edu.cn, hanchen.cui@sjtu.edu.cn,
{hangzhao,gaoyangiiis}@mail.tsinghua.edu.cn

**Abstract:** Robots rely heavily on sensors, especially RGB and depth cameras, to perceive and interact with the world. RGB cameras record 2D images with rich semantic information while missing precise spatial information. On the other side, depth cameras offer critical 3D geometry data but capture limited semantics. Therefore, integrating both modalities is crucial for learning representations for robotic perception and control. However, current research predominantly focuses on only one of these modalities, neglecting the benefits of incorporating both. To this end, we present **Semantic-Geometric Representation** (**SGR**), a universal perception module for robotics that leverages the rich semantic information of large-scale pre-trained 2D models and inherits the merits of 3D spatial reasoning. Our experiments demonstrate that SGR empowers the agent to successfully complete a diverse range of simulated and real-world robotic manipulation tasks, outperforming state-of-the-art methods significantly in both single-task and multi-task settings. Furthermore, SGR possesses the capability to generalize to novel semantic attributes, setting it apart from the other methods. Project website: semantic-geometric-representation.github.io.

**Keywords:** Representation Learning, Robotic Manipulation

## 1    Introduction

In the field of robotics, sensors play an indispensable role in enabling robots to perceive and interact with the world [1]. Among the various types of sensors, RGB cameras and depth cameras stand out for their accessibility and affordability [2]. RGB cameras capture high-resolution 2D images that are rich in semantic information, providing a high-level understanding of the scene [3, 4]. However, it is hard to use them to reason about complex spatial relationships. On the other hand, depth cameras provide 3D geometry information, which is crucial for accurate fine-grained manipulation [5, 6, 7]. Nevertheless, they have limited semantic understanding. As shown in Figure 1, it is of utmost importance in robot learning to utilize these two complementary modalities to develop general-purpose robots that can learn a diverse set of manipulation skills. In this paper, we ask the following question: *in the new era of pre-trained large vision models, how can we learn a representation for robots that integrates both semantic understanding and 3D spatial reasoning?*

A growing body of research is focused on utilizing pre-trained 2D vision models to learn visual representations that possess general knowledge of our world [8, 9, 10, 11, 12, 13]. This high-level semantic information is invaluable for generalizable robotic control. However, the current emphasis has largely been on pre-training on huge volumes of 2D images [14, 15], despite the fact that our world is intrinsically 3D. When confronted with the immense complexity of real-world environments, a robot with only 2D prior knowledge may face challenges such as partial occlusion

---

*Equal contribution

†Corresponding author

7th Conference on Robot Learning (CoRL 2023), Atlanta, USA.

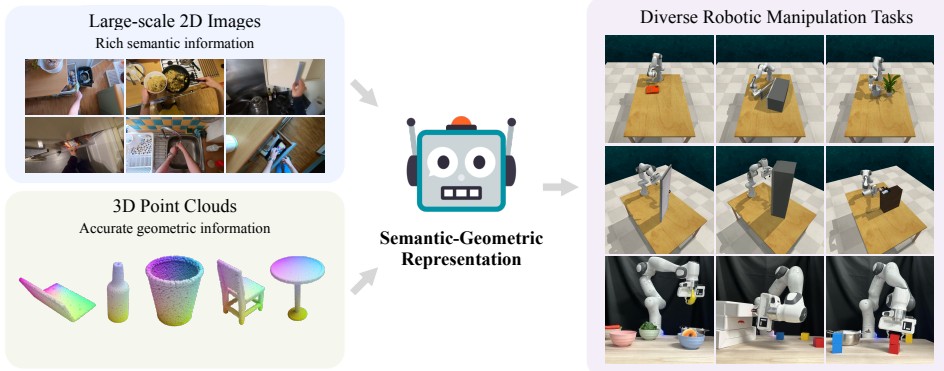

Figure 1: **Semantic-Geometric Representation (SGR):** Leveraging semantic information from massive 2D images and geometric information from 3D point clouds, we present a representation model SGR that enables the robots to solve a range of simulated and real-world manipulation tasks.

and geometric shape comprehension [16, 17]. Therefore, it is essential to develop models that can handle 3D visual data to enable robots to perform spatial reasoning effectively.

One natural and intuitive solution is to develop pre-trained 3D vision models. However, despite some attempts in the vision community [18, 19, 20, 21], current models are not yet sufficiently transferable to robotic manipulation tasks due to two main reasons. *(i)* The expensive data acquisition and labor-intensive annotation process result in a lack of large-scale 3D datasets [22, 23]. *(ii)* 3D point clouds with sparse structural patterns do not provide diversified semantics compared to colorful 2D images [24], thereby constraining the generalization capacity of 3D pre-training.

To address the challenges associated with using 2D or 3D vision models independently, we propose **Semantic-Geometric Representation (SGR)**, a hybrid perception module for robotics that captures both high-level semantics and low-level spatial patterns. As illustrated in Figure 2, we first utilize a large vision foundation model [25, 26, 27], pre-trained on massive amounts of internet data, to encode semantic feature maps from 2D images. Secondly, the context-rich 2D feature vectors are back-projected into 3D space and combined with the point cloud features that are extracted from point clouds using a point-based network [28, 29]. These fused features are fed into a number of set abstraction (SA) blocks [30], which jointly model the cross-modal interaction between 2D semantics and 3D geometry information. Finally, based on the output representations from the SA blocks, we predict the robotic action to execute.

We learn control policy through behavior cloning [31] on a broad spectrum of robotic manipulation tasks from the Robot Learning Benchmark (RLBench) [32]. In both single-task and multi-task learning scenarios, our SGR consistently outperforms a wide range of 2D and 3D representations, such as R3M [12], CLIP [26], and PERACT [33], by a significant margin. This striking performance demonstrates the effectiveness of integrating semantic and geometric representations synergistically. Furthermore, SGR stands as the approach capable of generalizing to previously unseen semantic attributes, including colors and shapes. To further validate its effectiveness, we assess SGR using a Franka Emika Panda robot, training a multi-task agent across 8 real-world tasks. Remarkably, SGR significantly surpasses a strong baseline (PERACT) on tasks that involve visual distractors. All these findings provide compelling evidence that SGR possesses the potential to serve as a universal representation for general-purpose robot learning.

## 2    Related Work

**2D Representation Learning for Robotics.** A wealth of prior approaches learn visual representations from *in-domain* data taken directly from the target environment and task. These techniques encompass a broad spectrum, ranging from data augmentation [34, 35, 36] and forward dynamics modeling [37, 38] to the utilization of task-specific information [39, 40]. However, recent studies

shift their focus towards acquiring more generalized representations. They achieve this by employing vision models pre-trained on large-scale *out-of-domain* data as the cornerstones for visuo-motor control [11, 41]. RRL [42], PIE-G [43], and MVP [10] demonstrating the effectiveness of supervised or self-supervised pre-trained representations for RL agents. PVR [9] and R3M [12] find that vision models pre-trained on real-world data enable data-efficient behavior cloning. VIP [44] proposes a pre-trained representation capable of producing dense reward signals. Hu et al. [8] conducts the first thorough empirical evaluation of pre-trained visual representations across different downstream policy learning methods. However, these approaches lack an explicit consideration of 3D geometry, which limits their capacity to develop highly accurate spatial manipulation skills in robotics.

**3D Representation Learning for Robotics.** The 3D perception module for robotics can be classified into the following three types: (1) Projection-based models: Many approaches utilize multi-view images, which represent the projections of a 3D environment onto various image planes, as direct inputs for robots [45, 46, 47]. However, a significant drawback of these methods is the loss of geometric information during the projection stage. (2) Voxel-based models: Another way to represent a 3D environment is voxelization, which extends the concept of 2D pixels [48, 49]. Recent works like C2FARM [50] and PERACT [33] employ a voxelized observation and action space for 6-DoF manipulation. Nevertheless, these models underutilize the sparsity of point sets in 3D and suffer from massive computational and memory costs. (3) Point-based models: In contrast to the previous two types, point-based models, such as PointNet [28], PointNet++ [30], directly and efficiently process point clouds. A plethora of approaches in robotics use PointNet or PointNet++ as a visual feature encoder [51, 52, 53, 54, 55, 56]. In this paper, we utilize point-based models to extract precise 3D geometric information from point clouds due to their effectiveness and efficiency.

**Fuse 2D and 3D information.** There is a rich body of literature dedicated to the study of sensor fusion, which aims to leverage the complementary nature of 2D and 3D information. In the field of autonomous driving, numerous previous works utilize camera-captured images and LiDAR point clouds to achieve accurate and robust 3D object detection [57, 58, 59, 60, 61]. Similarly, in the field of robotics, several prior studies explore the combination of color and depth information from RGB-D input to estimate object poses [62, 63, 64]. One notable approach, called DenseFusion [62], processes the two data sources individually and employs a fusion network to extract pixel-wise dense feature embeddings. However, one limitation of DenseFusion is its exclusive training on data from the target domain, leading to a failure in generalizing to novel objects. In contrast, our approach leverages visual representations pre-trained on large-scale images sourced from the web, empowering our robot to generalize to unseen attributes. This level of generalization is unattainable for a domain-specific method.

# 3 Method

In this section, we cover the different components of our approach. We start with a brief introduction of CLIP [26] and PointNeXt [28] in Section 3.1. Next, in Section 3.2, we present Semantic-Geometric Representation (SGR), a method that leverages the benefits of both pre-trained 2D models and 3D networks. Lastly, in Section 3.3, we demonstrate the integration of SGR into a behavior-cloning agent for 6-DoF manipulation tasks.

## 3.1 Background

**Contrastive Language-Image Pre-training (CLIP)** [26] has emerged as a simple yet powerful methodology for learning representations by pre-training on extensive image-text pairs from the web. Initially, CLIP employs separate modality-specific models to embed images and text. Subsequently, these vectors are projected into a shared embedding space and normalized. The InfoNCE loss [65] is calculated based on these final embeddings, utilizing corresponding images and captions as positive pairs and treating all non-matching images and captions as negative pairs. The learned visual representation not only exhibits impressive semantic discriminative power and zero-shot transferability [26], but also demonstrates exceptional generalization performance in robotic

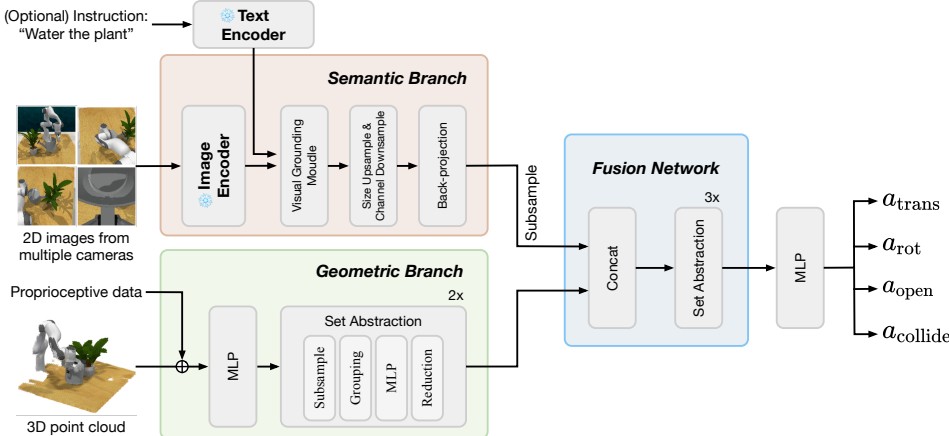

Figure 2: **Architecture.** The semantic branch generates point-wise semantic features from the multi-view images. Simultaneously, the geometric branch extracts geometric features from a point cloud. Finally, we fuse the two kinds of complementary features and predict the action to execute.

manipulation and navigation tasks [66, 67]. Although we utilize CLIP to extract semantic features, our perception module is also compatible with a wide range of pre-trained 2D vision models [68, 13].

**PointNet, PointNet++, and PointNeXt.** PointNet [28] is a pioneering and classical model that directly operates on unordered point clouds, capable of learning global point cloud features through shared multi-layer perceptrons (MLPs) and a max-pooling operation. PointNet++ [30] extends PointNet by utilizing a hierarchical neural network to handle local and global features in a more efficient manner. PointNeXt [29] is a recent study that revisits PointNet++ with improved training and scaling strategies to enhance its performance to the state-of-the-art level. Because of the simplicity, efficiency, and effectiveness of PointNeXt, we use PointNeXt in our experiments to enable robots to obtain accurate 3D spatial information.

## 3.2 Semantic-Geometric Representation

Our objective is to design a universal perception module for robotics that incorporates various sensory inputs, such as RGB images, point clouds, and proprioception data (e.g., gripper open state). This module generates a **Semantic-Geometric Representation (SGR)** capable of capturing both high-level semantics and low-level spatial patterns. The resulting representations are utilized to predict robotic actions for manipulation tasks. Moreover, for language-conditioned tasks, our module can process natural language and ground semantic concepts present in the images. The module comprises three key components: (1) a semantic branch, (2) a geometric branch, and (3) a fusion network. Figure 2 provides an overview of our design.

**Semantic branch.** Given a set of RGB images $\{I_i\}_{i=1}^K$ obtained from $K$ calibrated cameras, we first utilize a *frozen* pre-trained 2D model $\mathcal{G}$ (e.g., the visual encoder of CLIP) to extract multi-view image features $\{\mathcal{G}(I_i)\}_{i=1}^K$. If an language instruction $S$ is provided for a task, we employ a pre-trained language model $\mathcal{H}$ (e.g., the language encoder of CLIP) to obtain the language encoding $\mathcal{H}(S)$. Following MaskCLIP [69] and PROGRAMPORT [70], we use a visual grounding module to align each feature map $\mathcal{G}(I_i)$ with the language encoding $\mathcal{H}(S)$, resulting in aligned feature maps $\{M_i\}_{i=1}^K$. For more details on the visual grounding module, please refer to Appendix B.1. Next, we upsample the visual feature maps $\{\mathcal{G}(I_i)\}_{i=1}^K$ (or aligned feature maps $\{M_i\}_{i=1}^K$) to the same size as the input images by bilinear interpolation and downsample the channels via $1 \times 1$ convolution, resulting in a set of features denoted as $\{F_i\}_{i=1}^K$, where $F_i \in \mathbb{R}^{H \times W \times C_1}$ and $H, W$ represent the image size. These encoded 2D features, which encompass rich high-level semantics, are projected back into 3D space, yielding point-wise semantic features for the point cloud. We formulate them as $F^{3D} \in \mathbb{R}^{N \times C_1}$, where $N = K \times H \times W$.

**Geometric branch.** The raw point cloud, denoted as $P \in \mathbb{R}^{N \times 3}$, is generated by utilizing multi-view depth images and known camera parameters (i.e., camera intrinsics and extrinsics). Additionally, we incorporate robot proprioceptive information $z$ by applying a linear layer, which produces a $D$-dimensional vector that is attached to all the points. This augmented point cloud is represented as $P' \in \mathbb{R}^{N \times (3+D)}$. Subsequently, we employ a point-based network (e.g., a hierarchical PointNeXt) to process the point cloud $P'$. This processing step yields downsampled point geometric features, denoted as $G \in \mathbb{R}^{M \times C_2}$, where $M < N$. While these geometric features effectively capture local 3D structure and spatial characteristics, they lack the understanding of semantic information.

**Fusion network.** To integrate the two complementary branch, we begin by processing the point-wise semantic features $F^{\text{3D}}$ using the same point subsampling procedure as the geometric branch. The subsampled semantic features are denoted as $F_{\text{sub}}^{\text{3D}} \in \mathbb{R}^{M \times C_1}$. Next, we fuse the semantic features and geometric features through channel-wise concatenation, expressed as $F_{\text{fuse}}^{\text{3D}} = \text{Concat}(F_{\text{sub}}^{\text{3D}}, G) \in \mathbb{R}^{M \times (C_1 + C_2)}$. Finally, the fused features undergo several set abstraction blocks [30], enabling a cohesive modeling of the cross-modal interaction between 2D semantics and 3D geometric information. The resulting global feature represents our Semantic-Geometric Representation, which is utilized for action prediction. SGR back-projects 2D feature maps to 3D and fuses them with point cloud features directly in 3D space. This process fully utilizes the semantic information from pre-trained models and leads to better alignment with geometric information.

## 3.3 Robot Learning Framework

**Behavior Cloning.** Hu et al. [8] conduct a large-scale benchmarking study of pre-trained vision models using different policy learning methods. Their investigation show that evaluation based on reinforcement learning (RL) is highly variable and therefore unreliable, and further advocate for using more robust methods like behavior cloning (BC) [31]. Heeding this advice, we investigate the learning of 6-DoF manipulation tasks using BC. Inspired by PERACT [33], we formulate BC training as a task of predicting the "next best action". At each timestep, the robot receives an observation $O$ comprising multi-view RGB images $\{I_i\}_{i=1}^K$, organised point cloud $P$, and proprioceptive data $z$. Leveraging the perception module introduced in Section 3.2, the robot extracts the SGR representation and predicts the next best action $a$ in terms of target translation, rotation, and gripper state. Subsequently, this action is executed through a motion planner, and the process repeats until termination. Following James et al. [71, 50], all predicted robot actions are keyframe actions that capture bottleneck end-effector poses (i.e., joint velocities close to zero and unchanged gripper open state). This keyframe discovery strategy allows for the circumvention of direct predictions of long sequences of noisy actions, thus enabling more efficient policy learning. The robot is trained through supervised learning with input-keyframe action tuples from a dataset of expert demonstrations.

**Training Details.** Following the setup in PERACT [33], an action $a$ consists of the 6-DoF pose (translation and rotation), gripper open state, and whether the motion-planner used collision avoidance to reach an intermediate pose: $a = \{a_{\text{trans}}, a_{\text{rot}}, a_{\text{open}}, a_{\text{collide}}\}$. For rotations, the ground-truth action is represented as a one-hot vector per rotation axis with $R$ rotation bins: $a_{\text{rot}} \in \mathbb{R}^{(360/R) \times 3}$ ($R = 5$ degrees in our implementation). Open and collide actions are binary one-hot vectors: $a_{\text{open}} \in \mathbb{R}^2, a_{\text{collide}} \in \mathbb{R}^2$. The exception is translation, which is represented by a continuous 3D vector: $a_{\text{trans}} \in \mathbb{R}^3$. In this way, our loss objective is as follows:

$$\mathcal{L}_{\text{total}} = \lambda_1 \mathcal{L}_{\text{trans}} + \lambda_2 \mathcal{L}_{\text{rot}} + \lambda_3 \mathcal{L}_{\text{open}} + \lambda_4 \mathcal{L}_{\text{collide}}, \tag{1}$$

where $\mathcal{L}_{\text{trans}}$ is L1 loss for translation regression, and $\mathcal{L}_{\text{rot}}$, $\mathcal{L}_{\text{open}}$, and $\mathcal{L}_{\text{collide}}$ are cross-entropy losses for corresponding classification. Since these two kinds of losses have different scales, in our experiments, we set $\lambda_1 = 300$ and $\lambda_2 = \lambda_3 = \lambda_4 = 1$. We use AdamW [72] to optimize the model.

We utilize several data augmentation methods [29] to enhance the robustness and generalization of the model. During training, the translation is perturbed with $\pm 0.125$ m, and colors are randomly replaced with zero values, which are called translation perturbations and color drop respectively. Additionally, we adopt the point resampling strategy in both training and evaluation. For more details, we refer to Appendix B.2.

| Method | open microwave | open door | water plants | toilet seat up | phone on base | put books | take out umbrella | open fridge | **Average** |
|---|---|---|---|---|---|---|---|---|---|
| R3M | 7.0 | 24.2 | 1.2 | 59.7 | 0.0 | 15.3 | 28.3 | 4.7 | 17.6 |
| CLIP | 9.1 | 68.4 | 4.1 | 40.1 | 7.3 | 38.0 | 66.7 | 7.4 | 30.2 |
| PointNeXt$_{\text{ULIP}}$ | 9.4 | 60.0 | 10.4 | 63.0 | 35.6 | 41.8 | 49.8 | 15.3 | 35.7 |
| PointNeXt$_{\text{LfS}}$ | 17.0 | 63.8 | 28.2 | 52.7 | 65.9 | 53.1 | **95.1** | 7.3 | 47.9 |
| PERACT | 26.9 | 47.8 | **41.7** | 67.1 | 0.0 | 63.7 | 94.4 | 10.1 | 44.0 |
| **SGR (Ours)** | **52.6** | **80.2** | 40.2 | **80.1** | **81.1** | **84.8** | 94.7 | **34.8** | **68.6** |

Table 1: **Single-Task Test Results.** All numbers represent percentage success rates averaged over 3 seeds. See Appendix E for standard deviation. On average, our SGR outperforms PointNeXt$_{\text{LfS}}$, the most competitive baseline, with an improvement of $1.43\times$.

## 4 Experiments

Our experiments are designed to answer the following questions: (1) To what extent does SGR outperform methods exclusively concentrating on semantic or geometric information individually? (2) Is it feasible to train a multi-task model capable of handling all tasks, and how well does it perform? (3) Can SGR exhibit generalization capabilities to previously unseen semantic attributes such as colors and shapes? (4) How well does SGR perform on real-world tasks?

### 4.1 Simulation Setup

**Environment and Tasks.** Our simulated experiments are conducted on RLBench [32], a large-scale benchmark and learning environment designed for vision-guided manipulation research. The visual observations are obtained from four RGB-D cameras positioned at the front, left shoulder, right shoulder, and wrist of a Franka Emika Panda robot. All cameras have a resolution of $128 \times 128$ and are noiseless. We use 8 tasks with 100 demonstrations per task for training. For more details about individual tasks and language-conditioned tasks, please see Appendix A.

**Evaluation.** We employ a meticulous evaluation protocol to minimize variance in our results. Specifically, for every task, we train an agent for 40,000 iterations and save checkpoints every 1,000 iterations. Then we evaluate the last 20 checkpoints for 25 episodes and mark the top 3 best-performing checkpoints. Lastly, we evaluate the top 3 checkpoints for 100 episodes and get the average success rates as our test performances. The experiments are conducted with 3 seeds.

**Baselines.** To verify the effectiveness of SGR, we compare with the following baselines:
2D representations: We evaluate **CLIP** [26] and **R3M** [12]. Notably, R3M is pre-trained on diverse human video datasets and showcases remarkable performance on robotic manipulation tasks. For fair comparisons, we utilize frozen CLIP or R3M to process RGB images, employ a separate 2D CNN to process depth images, and subsequently fuse the two resulting features together [73].
3D representations: We examine **PERACT** [33], **PointNeXt$_{\text{LfS}}$** [29], and **PointNeXt$_{\text{ULIP}}$** [74]. PERACT is a voxel-based model that utilizes a Perceiver Transformer [75] to process voxelized observations. As for the point-based models, we consider both a Learning-from-Scratch PointNeXt (PointNeXt$_{\text{LfS}}$) and a frozen PointNeXt that has been pre-trained by ULIP [74] (PointNeXt$_{\text{ULIP}}$). ULIP is a recent 3D representation learning method that aligns features from images, text, and point clouds in the same space, following the philosophy of CLIP. Importantly, irrespective of whether 2D or 3D representations are employed, all the baselines utilize the same input modalities (RGB-D images) and the same multi-view (4 cameras) observations to ensure fair and meaningful comparisons.

### 4.2 Simulation Results

**Single-Task Performance.** Table 1 presents a comprehensive comparison of success rates achieved through single-task imitation learning using different representations. Initially, we observe a general underperformance of 2D representations compared to their 3D counterparts. This finding emphasizes the critical role of 3D geometric information in accomplishing fine-grained manipulation tasks. Among the 2D representations, CLIP demonstrates superior performance compared to R3M, suggesting that pre-trained R3M exhibits less transferability to the RLBench environment. Regard-

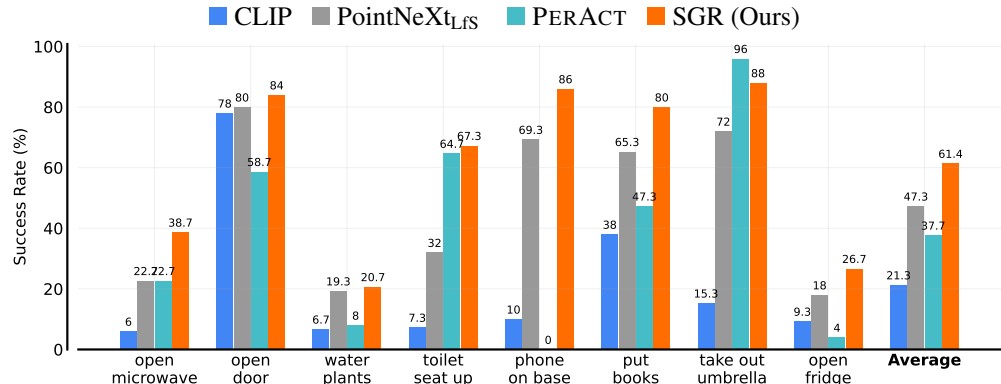

Figure 3: **Multi-Task Test Results.** We report the success rates averaged over 3 seeds. We observe that across 8 tasks SGR consistently outperforms baselines like CLIP, PointNeXt$_{\text{LfS}}$, and PERACT.

ing 3D representations, a Learning-from-Scratch PointNeXt (PointNeXt$_{\text{LfS}}$) outperforms its frozen counterpart pre-trained using ULIP (PointNeXt$_{\text{ULIP}}$). We hypothesize that this discrepancy arises due to the current focus of 3D pre-training methods on learning from small object-level datasets [22], resulting in representations that are less suitable for scene-level environments typically encountered in robotic manipulation tasks. Furthermore, PointNeXt$_{\text{LfS}}$ achieves comparable average results to PERACT while significantly reducing both computational and memory costs. Ultimately, our SGR proves to be the most effective representation, outperforming the second-best method by over 20% on average and yielding superior performance in 6 out of 8 tasks. These results highlight the benefits of capturing both 2D semantics and 3D spatial patterns for robotic manipulation. See more results about ablations and single-view comparisons in Appendix G and Appendix H, respectively.

**Multi-Task Performance.** In practical scenarios, our objective is to ensure that robots acquire a broad range of skills, rather than focusing on a single skill alone. With this goal in mind, we train a single agent on all 8 tasks, while also providing language instructions for each task to enhance the robot's ability to differentiate between them. For detailed information on the training of this language-conditioned behavior-cloning agent, please refer to the Appendix B.2. We compare our method, SGR, with three strong baselines and summarize the results in Figure 3. SGR demonstrates exceptional performance in multi-task scenarios, achieving an average accuracy of 61.4%. This outcome underscores the advantageous synergistic effect achieved by considering both semantic and geometric information, even in diverse multi-task settings. However, it is surprising to note that the performance of all multi-task models declines to varying extents when compared to their single-task counterparts. Notably, this phenomenon has also been observed in PERACT [33] and other recent studies [76, 77]. While multi-task training holds great promise for knowledge sharing among related tasks and facilitating more efficient learning, it presents numerous optimization challenges in practice [78, 79]. We hypothesize that employing advanced optimization algorithms [80, 81, 82] may help alleviate the negative transfer effects. We leave this exploration as future work.

**Generalizing to Novel Attributes.** A remarkable feature of human manipulation skill is that, once humans have learned to manipulate a category of objects, they are able to manipulate even unseen objects, regardless of their variations in color and shape. To evaluate whether an agent exhibits a similar level of generalization, we introduce three tasks: 1) reaching to a colored ball (`reach target`), 2) sliding a block onto a colored square target (`slide block`), and 3) reaching to a specified shape (`reach shape`). Notably, the `reach target` and `slide block` tasks examine the agent's ability to generalize across different colors, with training conducted on 10 colors and testing on 10 unseen colors. The `reach shape` task assesses the ability to generalize across various shapes, where training encompasses 3 shapes while testing involves 2 unseen shapes. Language instructions are employed to specify the desired color or shape. See Appendix A for more task details.

Figure 4 demonstrates that SGR achieves significantly higher performance when confronted with *unseen* colors or shapes, highlighting its remarkable generalization ability. This exceptional capacity primarily stems from the semantic branch of SGR (a pre-trained CLIP), which has undergone

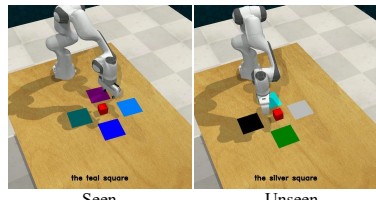

| Method | reach target Seen | reach target Unseen | slide block Seen | slide block Unseen | reach shape Seen | reach shape Unseen |
|---|---|---|---|---|---|---|
| R3M | 20.2 | 5.5 | 11.0 | 3.0 | 78.1 | 6.7 |
| CLIP | 42.0 | 17.7 | 49.0 | 16.3 | 95.5 | 0.2 |
| PointNeXt$_{ULIP}$ | - | - | - | - | 45.9 | 6.7 |
| PointNeXt$_{LfS}$ | 99.7 | 4.0 | 96.5 | 6.2 | 99.8 | 1.8 |
| DenseFusion | 71.8 | 9.5 | 26.5 | 16.5 | 99.3 | 3.8 |
| **SGR (Ours)** | 94.0 | **36.8** | 97.7 | **43.0** | 99.9 | **27.7** |

Figure 4: **Left:** For the `slide block` task, we train the agent on 10 colors and evaluate it on 10 unseen colors. Please see Appendix A for `reach target` and `reach shape` tasks. **Right:** Our SGR is the only method that demonstrates the ability to generalize across various colors and shapes. PointNeXt$_{ULIP}$ only accepts point coordinate input, thus the first two tasks are not applicable to it.

extensive exposure to a wide range of semantic attributes during its pre-training process on millions of image-caption pairs. In contrast, if we initialize the semantic branch randomly and train it from scratch, the resulting model would resemble DenseFusion [62]. As depicted in Figure 4, DenseFusion exhibits limited generalization ability. However, it is crucial to recognize that relying solely on semantic information proves inadequate. Even prominent pre-trained models such as CLIP and R3M encounter challenges in achieving satisfactory performance on the *seen* split of certain tasks, largely due to their inability to capture precise 3D spatial information.

### 4.3 Real-Robot Results

Finally, we evaluate the performance of SGR on real-world tasks. We design a set of 8 manipulation tasks using a Franka Emika Panda. See our website[3] for qualitative results that showcase the diversity of our tasks. We collect 8 demonstrations for each task and train a multi-task agent on all 8 tasks. Further details regarding the real robot setup can be found in Appendix D. In Table 2, we present the success rates and compare SGR with PERACT, one of the strong baselines we evaluated in simulation. SGR achieves an impressive average success rate of 59%, while PerAct attains 48%. However, it is important to note that our method is not flawless. The most common failure scenario involves predicting grasp poses that

| Success out of 10 Trials | SGR | PERACT |
|---|---|---|
| Hitting Ball | 70% | 30% |
| Putting Marker in Drawer | 70% | 60% |
| Moving Cup to Goal | 70% | 30% |
| Picking Red Block | 60% | 70% |
| Putting Banana in Pot | 60% | 60% |
| Putting Apple in Bowl | 50% | 70% |
| Opening Drawer | 50% | 30% |
| Pressing Handsan | 40% | 40% |
| Average | **59%** | 48% |

Table 2: Success rates of multi-task agents on real-world tasks.

are already in proximity to the object but lack sufficient precision. We speculate that this occurs because the distribution of human demonstrations is inherently multi-modal, and our translation action (which outputs a point estimate) tends to learn the average of this distribution. To address this limitation, future research could explore integrating SGR into methods that possess multi-modal modeling capabilities [83, 84].

## 5 Discussion

**Limitations.** Although we uncover the significance of harnessing both semantic and geometric information for robotic tasks, our current approach relies on modality-specific networks to extract these two complementary types of information and subsequently combines them using a straightforward method. However, we are thrilled by the prospect of employing a unified architecture, such as a multimodal Transformer [85], which would allow us to encode and fuse various modalities in a more elegant manner. This unified approach would also enable us to learn semantic-geometric representations during a pre-training phase, utilizing a wide range of data from real-world.

**Conclusion.** We present Semantic-Geometric Representation (SGR), a perception module for robotics that integrates high-level semantic understanding and low-level spatial reasoning. Through extensive evaluations in both single-task and multi-task learning scenarios, we observe remarkable performance improvements when employing SGR. Importantly, SGR exhibits an impressive ability to generalize to novel attributes, a pivotal factor in achieving general-purpose manipulation skills.

---

[3]`semantic-geometric-representation.github.io`

**Acknowledgments**

This work is supported by the Ministry of Science and Technology of the People's Republic of China, the 2030 Innovation Megaprojects "Program on New Generation Artificial Intelligence" (Grant No. 2021AAA0150000). This work is also supported by the National Key R&D Program of China (2022ZD0161700).

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

# A Task Details

**Setup.** Our simulation experiments are conducted on Robot Learning Benchmark (RLBench) environment [32].

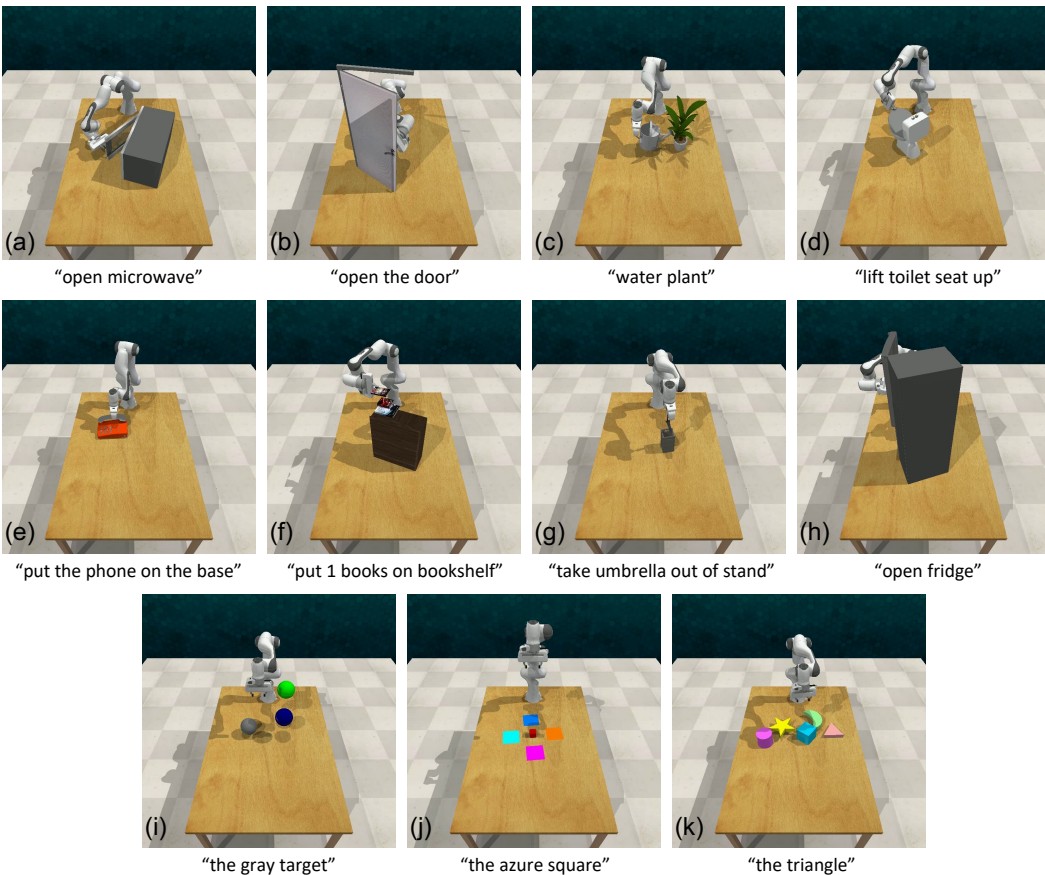

"open microwave"  "open the door"  "water plant"  "lift toilet seat up"

"put the phone on the base"  "put 1 books on bookshelf"  "take umbrella out of stand"  "open fridge"

"the gray target"  "the azure square"  "the triangle"

Figure 5: **RLBench Tasks.** Our simulation experiments encompass 11 tasks from RLBench. Tasks (a)-(h) serve as the basis for single-task and multi-task learning, while tasks (i)-(k) are employed to evaluate generalization performance.

For single-task and multi-task learning, we use 8 tasks that closely mirror real-world scenarios, for example, our tasks incorporate objects like microwaves, plants, phones, books, and toilets that align more closely with the reality of domestic robot environments. Our aim is to leverage the powerful semantic understanding and spatial reasoning capabilities of SGR to facilitate the manipulation of a diverse and complex array of objects in real-world situations.

For the generalization part, we conduct experiments on 3 tasks characterized by a substantial amount of visual variations. Among the 3 tasks, two focus on color generalization, while the remaining one is centered around shape generalization. The color variations include 20 instances: `colors` = {`red`, `maroon`, `lime`, `green`, `blue`, `navy`, `yellow`, `cyan`, `magenta`, `silver`, `gray`, `orange`, `olive`, `purple`, `teal`, `azure`, `violet`, `rose`, `black`, `white`}. The shape variations include 5 instances: `shapes` = {`cube`, `cylinder`, `triangle`, `star`, `moon`}.

As depicted in Figure 5, we utilize a diverse set of simulation tasks. In the following sections, we will provide a detailed examination of each task, and for those tasks that have been modified compared with the original RLBench implementation[4], we will discuss the specifics of the modifications in detail.

---

[4] https://github.com/stepjam/RLBench

### A.1 Open Microwave

**Filename:** `open_microwave.py`

**Task:** Open the microwave on the table.

**Modified:** No.

**Objects:** 1 microwave.

### A.2 Open Door

**Filename:** `open_door.py`

**Task:** Grip the handle and push the door open.

**Modified:** No.

**Objects:** 1 door.

### A.3 Water Plants

**Filename:** `water_plants.py`

**Task:** Pick up the watering can by its handle and water the plant.

**Modified:** No.

**Objects:** 1 watering can and 1 plant.

### A.4 Toilet Seat Up

**Filename:** `toilet_seat_up.py`

**Task:** Grip the edge of the toilet seat and lift it up to an upright position.

**Modified:** No.

**Objects:** 1 toilet with lid closed.

### A.5 Phone on Base

**Filename:** `phone_on_base.py`

**Task:** Grasp the phone and put it on the base.

**Modified:** No.

**Objects:** 1 phone and 1 phone base.

### A.6 Put Books

**Filename:** `put_books_on_bookshelf.py`

**Task:** Pick up a book and place them on the top shelf.

**Modified:** No. The original task has 3 variations, here we only use the first variation.

**Objects:** 1 bookshelf and 3 books.

### A.7 Take out Umbrella

**Filename:** `take_umbrella_out_of_umbrella_stand.py`

**Task:** Grasp the umbrella by its handle, lift it up and out of the stand.

**Modified:** No.

**Objects:** 1 umbrella and 1 umbrella stand.

## A.8 Open Fridge

**Filename:** `open_fridge.py`

**Task:** Grip the handle and slide the fridge door open.

**Modified:** No.

**Objects:** 1 fridge with door closed.

## A.9 Reach Target

**Filename:** `reach_target.py`

**Task:** Reach the language-instructed colored ball. There are 3 balls, distinguishable solely by their colors, which are randomly distributed in space. The 3 colors are sampled from the full set of 20 color instances, thus offering 20 variations based on the target color.

**Modified:** Yes. The sizes of the balls are enlarged so that they are distinguishable in the RGB-D input.

**Objects:** 3 balls.

## A.10 Slide Block

**Filename:** `slide_block_to_color_target.py`

**Task:** Slide the block to the language-instructed colored square targets. There are 3 distractor target of different colors. The 4 targets are placed symmetrically and the colors are sampled from the full set of 20 color instances, which makes the 20 variations.

**Modified:** Yes. The original `slide_block_to_target.py` task contains only one target. Three other targets are added.

**Objects:** 1 block and 4 colored target squares.

## A.11 Reach Shape

**Filename:** `reach_shape.py`

**Task:** Reach the language-instructed shape of the block. There are always 1 target shape and 4 distractor shapes, which are from the 5 shape instances and randomly placed on the table. The 5 different target shapes make the 5 variations.

**Modified:** Yes, newly added task.

**Objects:** 5 shapes.

# B  SGR Details

## B.1  Architecture Details

**Input Data.** The SGR model takes as input RGB images $\{I_i\}_{i=1}^{K}$ of size $H \times W$ and corresponding depth images of the same size from multiple camera views. Then point cloud, represented in the robot's base frame, is derived from depth images with known camera extrinsics and intrinsics. A key detail here is that the RGB images and the organised point cloud are aligned, ensuring a one-to-one correspondence between the points across the two data forms. In the simulation setting, we have $H = W = 128$ and $K = 4$, corresponding to four camera positions: front, left shoulder, right

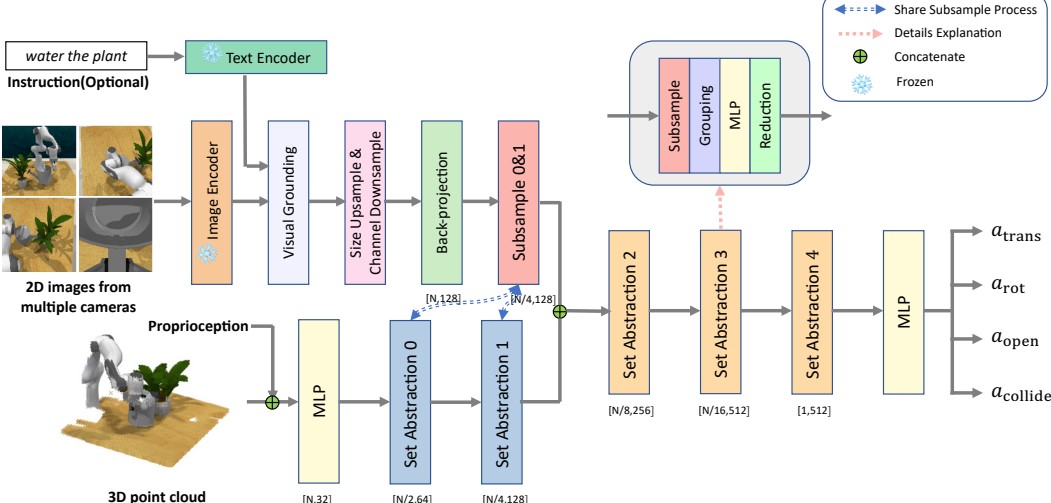

Figure 6: **Architecture Details.**

shoulder, and wrist. For real robot experiments, the setup is $H = W = 256$ and $K = 1$, denoting a single front-facing camera.

The model also receives proprioceptive data $z$, comprising four scalar values: gripper open state, left finger joint position, right finger joint position, and action sequence timestep. Additionally, if a task comes with language instruction $S$, this also forms part of the model's input.

**Visual Grounding Module.** The fundamental role of a visual grounding module is to localize visual areas that correspond to objects possessing a property as dictated by language. To equip our manipulation model with a deep comprehension of a variety of object properties and spatial relationships, including those not previously seen in the manipulation demonstration data, we take inspiration from MaskCLIP [69] and PROGRAMPORT [70].

It's worth noting that vanilla CLIP-ResNet-50 employs attention pooling to merge visual features from various spatial locations into one feature vector, and then uses a linear layer $\mathcal{F}$ as follows:

$$f_{\text{global}} = \mathcal{F}\left(\sum_i \text{softmax}\left(\frac{\text{Emb}_{\text{q}}(\text{AvgPool}(X)) \cdot \text{Emb}_{\text{k}}(X_i)^T}{C}\right) \text{Emb}_{\text{v}}(X_i)\right) \quad (2)$$

where $X_i$ is the input feature at spatial location $i$, $\text{Emb}_{q,k,v}(\cdot)$ are linear embedding layers and $C$ is some fixed scalar.

Following MaskCLIP [69] and PROGRAMPORT [70], we establish two $1 \times 1$ convolution layers, $\mathcal{C}_v$ and $\mathcal{C}_l$, which are initialized with the weights derived from $\text{Emb}_{\text{v}}$ and $\mathcal{F}$ respectively. Applying these layers in sequence enables the visual feature to be projected into the same space as the language feature. Ultimately, our grounding module formulates a dense mask encompassing image segments corresponding to the language $S$ in the following manner:

$$f_{\text{align}} = \sum \left(\text{TILE}(\mathcal{H}(S)) \odot \mathcal{C}_l(\mathcal{C}_v(X))\right) \in \mathbb{R}^{H' \times W' \times 1} \quad (3)$$

where $\mathcal{H}(S) \in \mathbb{R}^D$, $\mathcal{C}_l(\mathcal{C}_v(X)) \in \mathbb{R}^{H' \times W' \times D}$ ($H'$, $W'$, $D$ represent the height, width, and dimension of the pre-aligned feature respectively). $\mathcal{H}$ is the CLIP language encoder and TILE serves as a spatial broadcast operator that expands the language embedding vector to match the spatial dimensions of the visual features. The symbol $\odot$ indicates an element-wise multiplication, while the summation is performed over the feature dimension $D$. Additionally, we concatenate this feature with the pre-aligned image feature to generate our aligned feature maps:

$$M = \text{Concat}(f_{\text{align}}, \mathcal{C}_l(\mathcal{C}_v(X))) \in \mathbb{R}^{H' \times W' \times (1+D)} \quad (4)$$

**Other Details.** For the image encoder, we employ CLIP-ResNet-50. To maintain consistency with the resolution predominantly used in the pre-training of vision models, we upsample the input image size from $128 \times 128$ to $256 \times 256$ in simulation experiments. The CLIP text feature exhibits a dimensionality of 1024, and the CLIP image encoder generates a feature map of $8 \times 8$ with 2048 channels. For the 3D encoder, we utilize PointNeXt-S. Detailed information regarding the number of channels and sub-points in each layer of the PointNeXt network can be seen in Figure 6.

## B.2 Training Details

**Data Augmentation.** To further improve the performance and robustness of our model, we incorporate a variety of data augmentation techniques. (1) **Translation perturbations**: during training, the translation is perturbed with $\pm 0.125$ m. We also experimented with rotation perturbations but found they did not contribute to performance improvement. (2) **Color drop** is to randomly replace colors with zero values. This technique serves as a powerful augmentation for PointNeXt, leading to significant enhancements in the performance of tasks where color information is available. Qian et al. [29] suggest that by implementing color drop, the network is encouraged to pay more attention to the geometric relationships between points, leading to a boost in overall performance. (3) **Point resampling** is a common technique utilized in the processing of point cloud data, often applied to adjust the density of the point cloud. The technique involves selecting a subset of points from the original data, thereby creating a new dataset with altered density. In our simulation experiment, we resample 4096 points from the original point cloud. (4) **Keyframe discovery and demo augmentation** [50] [33] are to convert every data point within a demonstration into a task that involves predicting the subsequent next-best keyframe action. Keyframes are extracted by a heuristic method: an action is a keyframe if the joint velocities are close to zero and the gripper open state is unchanged. This keyframe discovery strategy bypasses the need to predict long sequences of noisy actions, making policy learning more efficient.

**Hyperparameters.** The configuration of hyperparameters applied in our studies can be found in Table 3. The single-task learning and generalization experiments are trained on a single NVIDIA GeForce RTX 3090 GPU, while the multi-task learning is trained on 8 GPUs. Since color plays a crucial role in some tasks and disregarding color information would render these tasks unsolvable, such as `reach target` and `slide block`, the color drop is not utilized in the generalization experiments.

Table 3: Hyper-parameters used in our simulation experiments.

| Config | Single-task | Multi-task | Generalization |
|---|---|---|---|
| Training iterations | $40,000$ | $40,000$ | $40,000$ |
| Leraning rate | $0.0003$ | $0.001$ | $0.0003$ |
| Batch size | $32$ | $256$ | $32$ |
| Optimizer | AdamW | AdamW | AdamW |
| Weight decay | $1 \times 10^{-6}$ | $1 \times 10^{-6}$ | $1 \times 10^{-6}$ |
| Color drop | $0.2$ | $0.4$ | $0$ |
| Instruction provided | No | Yes | Yes |
| Number of input points | $4096$ | $4096$ | $4096$ |

## C   Evaluation Workflow

**Simulation.** In pursuit of the reliability and stability of our results, our evaluation follows such subsequent steps. (1) Train the agent on the train set for 40,000 training iterations and save checkpoints every 1,000 iterations. (2) Evaluate the last 20 checkpoints for 25 episodes. (3) Evaluate the best 3 checkpoints for 100 episodes, and use the average performance of the 3 checkpoints as the results. (4) Repeat the above steps for 3 random seeds.

| Method | open microwave | open door | water plants | toilet seat up | phone on base | put books | take out umbrella | open fridge | Average |
|---|---|---|---|---|---|---|---|---|---|
| R3M | $7.0 \pm 5.9$ | $24.2 \pm 10.9$ | $1.2 \pm 0.7$ | $59.7 \pm 21.7$ | $0.0 \pm 0.0$ | $15.3 \pm 16.5$ | $28.3 \pm 19.0$ | $4.7 \pm 3.8$ | 17.6 |
| CLIP | $9.1 \pm 10.8$ | $68.4 \pm 8.8$ | $4.1 \pm 1.8$ | $40.1 \pm 20.8$ | $7.3 \pm 2.5$ | $38.0 \pm 16.3$ | $66.7 \pm 8.2$ | $7.4 \pm 2.8$ | 30.2 |
| PointNeXt$_{\text{ULIP}}$ | $9.4 \pm 1.6$ | $60.0 \pm 5.5$ | $10.4 \pm 1.6$ | $63.0 \pm 7.9$ | $35.6 \pm 4.8$ | $41.8 \pm 1.7$ | $49.8 \pm 4.2$ | $15.3 \pm 1.2$ | 35.7 |
| PointNeXt$_{\text{LfS}}$ | $17.0 \pm 5.4$ | $63.8 \pm 6.2$ | $28.2 \pm 2.2$ | $52.7 \pm 14.5$ | $65.9 \pm 19.9$ | $53.1 \pm 16.0$ | $\mathbf{95.1 \pm 3.0}$ | $7.3 \pm 4.7$ | 47.9 |
| PERACT | $26.9 \pm 4.3$ | $47.8 \pm 41.1$ | $\mathbf{41.7 \pm 3.1}$ | $67.1 \pm 4.2$ | $0.0 \pm 0.0$ | $63.7 \pm 3.5$ | $94.4 \pm 1.3$ | $10.1 \pm 3.9$ | 44.0 |
| **SGR (Ours)** | $\mathbf{52.6 \pm 5.1}$ | $\mathbf{80.2 \pm 2.7}$ | $40.2 \pm 8.3$ | $\mathbf{80.1 \pm 1.2}$ | $\mathbf{81.1 \pm 5.7}$ | $\mathbf{84.8 \pm 7.4}$ | $94.7 \pm 2.3$ | $\mathbf{34.8 \pm 2.0}$ | **68.6** |

Table 4: Single-task results with mean and standard deviation (%).

| Method | open microwave | open door | water plants | toilet seat up | phone on base | put books | take out umbrella | open fridge | Average |
|---|---|---|---|---|---|---|---|---|---|
| CLIP | $6.0 \pm 6.9$ | $78.0 \pm 2.0$ | $6.7 \pm 6.1$ | $7.3 \pm 8.1$ | $10.0 \pm 10.4$ | $38.0 \pm 21.2$ | $15.3 \pm 13.3$ | $9.3 \pm 5.0$ | 21.3 |
| PointNeXt$_{\text{LfS}}$ | $22.7 \pm 1.2$ | $80.0 \pm 11.1$ | $19.3 \pm 3.1$ | $32.0 \pm 20.0$ | $69.3 \pm 12.2$ | $65.3 \pm 13.3$ | $72.0 \pm 10.0$ | $18.0 \pm 2.0$ | 47.3 |
| PERACT | $22.7 \pm 2.3$ | $58.7 \pm 11.4$ | $8.0 \pm 10.6$ | $64.7 \pm 12.9$ | $0.0 \pm 0.0$ | $47.3 \pm 18.1$ | $\mathbf{96.0 \pm 2.0}$ | $4.0 \pm 3.5$ | 37.7 |
| **SGR (Ours)** | $\mathbf{38.7 \pm 15.0}$ | $\mathbf{84.0 \pm 3.5}$ | $\mathbf{20.7 \pm 1.2}$ | $\mathbf{67.3 \pm 10.3}$ | $\mathbf{86.0 \pm 10.6}$ | $\mathbf{80.0 \pm 15.6}$ | $88.0 \pm 6.9$ | $\mathbf{26.7 \pm 11.7}$ | **61.4** |

Table 5: Multi-task results with mean and standard deviation (%).

**Real-Robot.** For the real-robot experiments, we train the agent for 40,000 iterations and simply choose the last checkpoint for evaluation, since it's expensive to evaluate more checkpoints in the real robot. We evaluate 10 episodes per task. During the evaluation of a trained agent, the agent keeps acting until achieving the goal or reaching the maximum episode length.

## D    Real-Robot Setup

For our real-robot experiments, we use a Franka Emika Panda manipulator equipped with a parallel gripper. Perception is achieved through an Intel RealSense D415 camera, positioned in front of the scene. The camera generates RGB-D images with a resolution of $1280 \times 720$. We leverage the `realsense-ros`[5] to align depth images with color images. The extrinsic calibration between the camera frame and robot base frame is carried out using the `easy_handeye` package[6].

When preprocessing the RGB-D images, we initially crop the $1280 \times 720$ images to a $720 \times 720$ frame, and then resize them to $256 \times 256$ using nearest-neighbor interpolation. This interpolation is preferred over others, such as bilinear interpolation, as the latter can cause non-existent points in the depth map, leading to a noisy point cloud. Following these steps, we can process RGB-D images in the same manner as in our simulation experiments. It's important to note that the camera's intrinsics must be adjusted accordingly after the images are cropped and resized. We train SGR for 40,000 training steps with 64 demonstrations in total and use the final checkpoint for inference.

## E    Additional Results

For all simulation experiments, we use 3 random seeds to ensure the reliability of our results. While we present averaged results in the main body of the paper for clarity, we provide more comprehensive results in this section. Here, we report both the mean and standard deviation derived from our simulation results to offer a complete view of our experimental outcomes. Table 4, Table 5, and Table 6 show the results of single-task learning, multi-task learning, and generalization, respectively.

## F    Computational Efficiency

SGR stands out significantly in computational efficiency, boasting impressive gains in both memory and time efficiency. In the single-task setting, SGR converges within 7 hours using a batch size of 32 and consumes a mere 17GB of GPU memory. This training can be efficiently executed on a single NVIDIA RTX 3090 GPU. In comparison, the voxel-based method, PerAct, can only accommodate a batch size of 2 on the same GPU and requires over 16 hours to converge.

---

[5] https://github.com/IntelRealSense/realsense-ros
[6] https://github.com/IFL-CAMP/easy_handeye

| | reach target | | slide block | | reach shape | |
|---|---|---|---|---|---|---|
| Method | Seen | Unseen | Seen | Unseen | Seen | Unseen |
| R3M | $20.2 \pm 6.2$ | $5.5 \pm 0.9$ | $11.0 \pm 5.9$ | $3.0 \pm 2.8$ | $78.1 \pm 24.5$ | $6.7 \pm 8.2$ |
| CLIP | $42.0 \pm 27.0$ | $17.7 \pm 4.8$ | $49.0 \pm 14.3$ | $16.3 \pm 2.8$ | $95.5 \pm 2.8$ | $0.2 \pm 0.3$ |
| PointNeXt$_{ULIP}$ | - | - | - | - | $45.9 \pm 2.0$ | $6.7 \pm 4.7$ |
| PointNeXt$_{LfS}$ | $99.7 \pm 0.3$ | $4.0 \pm 2.3$ | $96.5 \pm 1.0$ | $6.2 \pm 0.8$ | $99.8 \pm 0.4$ | $1.8 \pm 1.6$ |
| DenseFusion | $71.8 \pm 4.0$ | $9.5 \pm 3.1$ | $26.5 \pm 1.0$ | $16.5 \pm 1.5$ | $99.3 \pm 0.6$ | $3.8 \pm 3.4$ |
| **SGR (Ours)** | $94.0 \pm 2.6$ | $\mathbf{36.8 \pm 1.6}$ | $97.7 \pm 1.3$ | $\mathbf{43.0 \pm 3.0}$ | $99.9 \pm 0.2$ | $\mathbf{27.7 \pm 4.0}$ |

Table 6: Generalization results with mean and standard deviation (%) .

| Method | open microwave | open door | water plants | toilet seat up | phone on base | put books | take out umbrella | open fridge | **Average** |
|---|---|---|---|---|---|---|---|---|---|
| SGR | 52.6 | 80.2 | 40.2 | 80.1 | 81.1 | 84.8 | 94.7 | 34.8 | 68.6 |
| SGR w/ RGBs added for points | 37.0 | 71.8 | 28.4 | 44.9 | 77.2 | 61.7 | 88.4 | 4.7 | 51.7 |
| SGR w/o semantic branch (PointNeXt$_{LfS}$) | 17.0 | 63.8 | 28.2 | 52.7 | 65.9 | 53.1 | 95.1 | 7.3 | 47.9 |
| SGR w/o geometric branch (CLIP) | 9.1 | 68.4 | 4.1 | 40.1 | 7.3 | 38.0 | 66.7 | 7.4 | 30.2 |
| SGR LfS (DenseFusion) | 20.4 | 87.5 | 31.5 | 80.2 | 64.2 | 63.9 | 75.7 | 25.0 | 56.0 |

Table 7: Ablation experiments on single-task setting.

The noticeable disparity in efficiency stems mainly from SGR's design. Its semantic branch uses a frozen pre-trained 2D model, while the geometric branch adopts point-based models. These point-based models exploit the sparsity of 3D point sets and inherently demand fewer computational resources than their voxel-based counterparts.

# G   Ablation Experiments

In order to delve deeper into the understanding and evaluation of the proposed method, we conduct a series of ablation experiments. The results are presented in Table 7.

**Adding RGB colors for each point in the geometric branch.** We conduct experiments with adding RGB colors for each point in the geometric branch. Notably, this addition led to a reduction in performance. We hypothesize that this decline in performance could be attributed to the lack of pre-training on colored point clouds, which might render the geometric branch susceptible to overfitting certain spurious RGB features.

**Deleting one of the two branches.** Indeed, certain baseline models outlined in Section 4.1 can be interpreted as deleting one of the two branches in our SGR model. In particular, PointNeXt$_{LfS}$ and CLIP are designed to reflect scenarios where only the geometric or semantic branch is retained, respectively. Additionally, to ensure a fair comparison, we make sure all ablation models utilize the same input modalities (RGB-D images). In the case of PointNeXt$_{LfS}$, we add RGB colors into the point cloud. For the CLIP model, we employ a separate 2D CNN to process depth images and subsequently fuse the CLIP features and depth features together.

**Switching out the pre-trained encoder with one trained from scratch.** This aligns with the implementation of DenseFusion in Section 4.2. The results presented in Figure 4 demonstrate that replacing a pre-trained encoder with one that is trained from scratch will significantly reduce its generalization capabilities. Furthermore, to offer a more comprehensive perspective, we have supplemented with experimental results in Table 7 for DenseFusion under the single-task setting. The results also demonstrate the significant role of the pre-trained model within SGR.

# H   Single-view Setup

In Section 4.1, all simulation experiments are conducted under a multi-view setup with 4 cameras. To more comprehensively demonstrate the advantages of SGR, we undertake further experiments to compare the performance of SGR, CLIP, and R3M under a single-view setup, utilizing the front camera. These experiments are executed in a single-task setting and the results are presented in Table 8.

| Method | open microwave | open door | water plants | toilet seat up | phone on base | put books | take out umbrella | open fridge | Average |
|---|---|---|---|---|---|---|---|---|---|
| R3M | 21.1 | 57.0 | 19.8 | 34.3 | 5.2 | 50.9 | 51.4 | 12.4 | 31.5 |
| CLIP | 18.2 | 48.6 | **25.5** | 61.3 | 20.8 | 27.9 | 28.2 | 7.0 | 29.7 |
| **SGR (Ours)** | **31.2** | **81.3** | 23.1 | **65.0** | **70.5** | **64.1** | **88.0** | **18.9** | **55.3** |

Table 8: Single-view (the front camera) performance under single-task setup.

We note that, compared with the multi-view setting in Table 1, SGR's performance experiences a reduction (from 68.6% to 55.3%). This decline is largely attributed to the fact that a single-view camera provides limited information, which in turn weakens the geometric understanding capability of SGR's geometric branch. Meanwhile, CLIP maintains a consistent performance (30.2% vs. 29.7%), and interestingly, R3M even witnesses an improvement (from 17.6% to 31.5%). We hypothesize that this phenomenon occurs because both CLIP and R3M were originally pre-trained in a single-view setup. As a result, the multi-view setting might not provide additional benefits for their 2D representations and might even introduce some interference.

Nonetheless, it's important to note that even in the single-view setting, owing to the spatial relationship captured by the geometric branch, SGR still stands out and exhibits superior performance compared to CLIP and R3M.

# I  Additional Related Work

**Generalization in Robotics.** Generalization is a pivotal foundation for the extensive application of robots in the real world. Some previous works [86] [87] have tackled challenging category-level manipulation tasks and demonstrated robust generalization in real-world scenarios. However, both works heavily rely on human priors, including manual specification of semantic keypoints [86] and the use of NUNOCS Net [87]. Such dependencies, coupled with their extensive and intricate manipulation pipelines, can pose scalability challenges in diverse tasks and environments. Contrarily, our approach focuses on learning a simple and scalable representation for visuo-motor control. This end-to-end approach is not only more efficient but also facilitates ease of implementation and deployment.

