# OpenReview forum: "A Universal Semantic-Geometric Representation for Robotic Manipulation"
_robot-learning.org/CoRL/2023/Conference — CoRL 2023 Poster_

### Official Review · Reviewer_e9U4 · 2023-07-16

**Confidence:** 3
**Originality:** Very Good
**Technical Quality:** Very Good
**Clarity Of Presentation:** Excellent
**Impact:** 2

**Recommendation:**

Strong Accept: I recommend accepting the paper and will argue for my recommendation even if other reviewers hold a different opinion.

**Review:**

The paper is very clear and follows a logical flow making it a pleasure to read. The idea of combining RGB information with depth is natural and the way this paper adopt is innovative.

The literature review is comprehensive. The reviewer especially likes the style of splitting into different category and at end of each highlighting the difference between this and aforementioned works.

The architecture image in Figure 2 is informative and clearly shows the framework.

The new representation (SGR) is then used with behavior cloning (BC) and compared with multiple SOTA and shows significant improvement in most of the tasks.


**Quality Of The Limitations Section:**

Limitations are addressed clearly

**Questions For Rebuttal:**

I in general in favor of accepting the paper therefore I don’t have further questions for rebuttal.

**Robotics Focus:**

Sufficient demonstration on hardware

**Summary Of Paper:**

This paper proposed a way to fuse semantic information with RGB data and spatial information with depth data to generate better representation for robot manipulation.

**Summary Of Recommendation:**

Overall I found the paper of very good quality, the idea is innovative and supported with both simulation & real robotics experiments, thusI recommend accepting the paper.

---

> ### Author Response · Authors · 2023-08-11
> **Response to Reviewer e9U4**
>
> Thank you for your positive feedback. We appreciate your recognition of our paper's clarity, innovative approach, and thorough literature review. Your comments about Figure 2 and our SGR representation's performance are especially encouraging. We're grateful for your kind words and will continue to strive for excellence in our research.

---

### Official Review · Reviewer_B5ZP · 2023-07-17

**Confidence:** 3
**Originality:** Good
**Technical Quality:** Good
**Clarity Of Presentation:** Good
**Impact:** 3

**Recommendation:**

Weak Reject: I recommend rejecting the paper, but will not argue for my recommendation if the majority of other reviewers have a different opinion.

**Review:**

Strengths:
- Despite some clarifications missing as mentioned below, the paper overall is well-written and easy to follow

- The idea of fusing semantic and geometric information is reasonable. The pipeline is appropriately designed.

- The experiments in both simulation and real world have been conducted. The results are encouraging and outperforms the baselines by a large margin.


Weakness and questions:
- In experiments, the proposed method leverages 4 cameras for observation. However, for comparison methods (both 2D and 3D), it is not clearly mentioned whether the same multi-view observation has been provided for fair comparison. Also some of the methods were not pretrained using multi-view images and thus cannot take full advantage of the multi-view. To solely validate the proposed fusion technique, single-view setup comparison would be helpful.

- In intro, the statement of “SGR stands as the only approach capable of generalizing to previously unseen semantic attributes, including colors and shapes” seems over-claimed. In fact, multiple prior works [A-D] have tackled much more challenging category-level manipulation tasks in the real world. Discussion or comparison to these line of works is encouraged to be added into the paper.

- It is mentioned “robot proprioceptive information is incorporated”. However, this is not reflected in fig2. It is suggested to ensure the consistency.

- It would be more interesting to show side-by-side comparison to the baselines in the qualitative results or in the video demos.

Overall, the work is potentially suitable for publication but needs some more clarifications given the current state. Addressing the above concerns is highly encouraged.

References:
    A) Manuelli, L., Gao, W., Florence, P. and Tedrake, R., 2019, October. kpam: Keypoint affordances for category-level robotic manipulation. In The International Symposium of Robotics Research (pp. 132-157). Cham: Springer International Publishing.
    B) Wen, B., Lian, W., Bekris, K. and Schaal, S., 2022. You only demonstrate once: Category-level manipulation from single visual demonstration. RSS 2022

**Quality Of The Limitations Section:**

Limitations are addressed clearly

**Questions For Rebuttal:**

see above

**Robotics Focus:**

Sufficient demonstration on hardware

**Summary Of Paper:**

This work presents Semantic-Geometric Representation (SGR), a universal perception module for robotics that leverages the rich semantic information of large-scale pre-trained 2D models and inherits the merits of 3D spatial reasoning. To do so, a network is designed to fuse the information from semantic and geometric features from 3D an 2D modalities respectively.


**Summary Of Recommendation:**

Overall, the work is potentially suitable for publication but needs some more clarifications given the current state. Addressing the above concerns is highly encouraged.

---

> ### Author Response · Authors · 2023-08-11
> **Response to Reviewer B5ZP - Part 1**
>
> We thank Reviewer B5ZP for their thorough and insightful review.
>
> > In experiments, the proposed method leverages 4 cameras for observation. However, for comparison methods (both 2D and 3D), it is not clearly mentioned whether the same multi-view observation has been provided for fair comparison.
>
> Thank you for bringing this to our attention. Indeed, for the sake of a fair comparison, all the baselines we evaluated utilized the same multi-view observations derived from the 4 cameras. We will clarify this point explicitly in the revised manuscript.
>
> > Also some of the methods were not pretrained using multi-view images and thus cannot take full advantage of the multi-view. To solely validate the proposed fusion technique, single-view setup comparison would be helpful.
>
> Thank you for raising this concern. We have undertaken further experiments to compare the performance of SGR, CLIP, and R3M under a **single-view setup**, utilizing the front camera. These experiments were executed in a single-task setting and the results are presented in the following table.
>
> We note that SGR's performance experiences a reduction (from 68.6% to 55.3%). This decline is largely attributed to the fact that a single-view camera provides limited information, which in turn weakens the geometric understanding capability of SGR's geometric branch. Meanwhile, CLIP maintains a consistent performance (30.2% vs. 29.7%), and interestingly, R3M even witnesses an improvement (from 17.6% to 31.5%). We hypothesize that this phenomenon occurs because both CLIP and R3M were originally pre-trained in a single-view setup. As a result, the multi-view setting might not provide additional benefits for their 2D representations and might even introduce some interference. This indeed validates your question, and we truly appreciate you pointing it out.
>
> In light of this insight, we intend to incorporate an explanatory section within the paper, dedicated to elucidating this topic. Nonetheless, it's important to note that even in the single-view setting, owing to the spatial relationship captured by the geometric branch, SGR still stands out and exhibits superior performance compared to CLIP and R3M.
>
> |                | open microwave | open door | water plants | toilet seat up | phone on base | put books | take out umbrella | open fridge | **Average**  |
> |:--------------:|:--------------:|:---------:|:------------:|:--------------:|:-------------:|:---------:|:-----------------:|:-----------:|:---------:|
> | **SGR**        |      31.2      |   81.3    |     23.1     |      65.0      |      70.5     |   64.1    |        88.0       |     18.9    |    55.3   |
> | **CLIP**       |      18.2      |   48.6    |     25.5     |      61.3      |      20.8     |   27.9    |        28.2       |      7.0    |    29.7   |
> | **R3M**        |      21.1      |   57.0    |     19.8     |      34.3      |       5.2     |   50.9    |        51.4       |     12.4    |    31.5   |

---

### Official Review · Reviewer_ebNV · 2023-07-20

**Confidence:** 3
**Originality:** Good
**Technical Quality:** Good
**Clarity Of Presentation:** Good
**Impact:** 3

**Recommendation:**

Weak Reject: I recommend rejecting the paper, but will not argue for my recommendation if the majority of other reviewers have a different opinion.

**Review:**

**Strengths.**

* The paper tackles an important problem of constructing a universal visual representation for robot manipulation. The proposed method is well-motivated and clearly presented.
* The idea of making use of both 2D and 3D representation learning methods for the purpose of robot manipulation seems to be an under-explored area.
* The authors conducted extensive experiments and showed quantitative results including comparisons with baselines in both sim and real.

**Clarification questions with the proposed method.**

* What is the benefit of processing 2D images and 3D point clouds in two branches when one could design a framework that makes use of pre-trained vision models (eg. CLIP) while processing both 2D images and 3D point clouds at the same time? For example, one could take a PerAct-like approach to use a transformer-based architecture, process the 2D images with CLIP-based visual encoders and feed the encoded representations into the transformer as additional inputs.
* In the semantic branch, it seems that only RGB images from multiple views are used as inputs. Could the model benefit from processing the depth images from multiple views too?
* In the geometric branch, why is only the raw point cloud loaded, when the RGB colors of the points could be easily added for each point?

**Issues with experiments.** Although the authors performed a decent range of experiments on both simulation and real robot platforms, several problems exist in the details of those experiments and need to be addressed.

* The authors performed sim experiments on 8 RLBench tasks, but it seems that those tasks do not overlap the 18 RLBench tasks PerAct evaluated on. Could the authors pick a few tasks evaluated in PerAct and show at least some single-task quantitative results on them?
* In the "generalizing to novel attributes" experiments presented in Figure 4, the PerAct baseline is missing. Is there a particular reason results on this baseline are not presented? It would be nice if the authors could run and present results on PerAct in this table.
* It seems that the quantitative results of the real robot experiments are not very good. There indeed seems to be a gap between the performance of SGR and PerAct. However, out of the 8 tasks evaluated, SGR only outperforms PerAct in 4 of them.
* The paper lacks ablation experiments that study the extent to which parts of the method contribute to the overall performance. Some experiments that the authors could consider performing include (1) deleting one of the two branches in the network architecture while keeping other components of the framework; (2) switching out the pre-trained encoder with one that is trained from scratch.

**Quality Of The Limitations Section:**

Limitations are addressed clearly

**Questions For Rebuttal:**

* Justify design choices in the proposed method according to the clarification questions listed above.
* Add additional simulation experiments on RLBench tasks shared with the PerAct paper.
* Add PerAct baseline in Figure 4.
* Add ablation experiments if the authors also believe they add value to the paper.

**Robotics Focus:**

Sufficient demonstration on hardware

**Summary Of Paper:**

This paper presents a robot perception framework that processes 2D images and point cloud information in two separate branches. The 2D sensor images are processed in a semantic branch equipped with pre-trained visual encoders so that the model could extract relevant semantic information in the scene. The raw point cloud is processed in the geometric branch with a point-based network so that the encoded representation in this branch captures the 3D structure of the scene. The outputs of the two branches are integrated into a fusion network to produce a global feature, which is then used to predict the next robot action.

The paper presents results on using the proposed framework in behavior cloning in 8 simulation environments and 8 real robot tasks. The results show that the proposed method outperforms various baselines that use different 2D representations or 3D representations.

**Summary Of Recommendation:**

Overall, this is a nice paper with an interesting idea. However, because of the existing issues in the experiments section, my internal rating of this paper is borderline. I would be happy to adjust my rating if the authors make their experimental results more convincing.

---

> ### Author Response · Authors · 2023-08-11
> **Response to Reviewer ebNV - Part 1**
>
>
> We thank Reviewer ebNV for their thorough and insightful review.
>
> > What is the benefit of processing 2D images and 3D point clouds in two branches when one could design a framework that makes use of pre-trained vision models (eg. CLIP) while processing both 2D images and 3D point clouds at the same time? For example, one could take a PerAct-like approach to use a transformer-based architecture, process the 2D images with CLIP-based visual encoders and feed the encoded representations into the transformer as additional inputs.
>
> Thank you for highlighting this point, which offers an intriguing perspective on how 2D images and 3D point clouds could be processed. The reasons we process 2D images and 3D point clouds in two separate branches are as follows:
> - **Computational Efficiency**: PerAct-like approach leverages a transformer-based model to process over an extensive number of voxels. However, this comes at the cost of substantial computational and memory demands. In contrast, our approach exhibits remarkable efficiency in both memory utilization and training time. In the single-task setting, SGR converges within 7 hours using a batch size of 32 and consumes a mere 17GB of GPU memory. In comparison, PerAct can only accommodate a batch size of 2 on the same GPU and requires over 16 hours to converge. Integrating CLIP-based visual encoders into PerAct would exacerbate the already considerable computational burden.
> - **Direct 3D Space Mapping**: The approach you've suggested relies on transformers to implicitly learn the correspondence between 2D image features and 3D voxels. In contrast, our method employs a more direct approach by explicitly mapping these modalities in the 3D space. We believe that such explicit mapping can provide a more straightforward and efficient fusion of the two modalities.
> - **Challenges of Designing a Unified Network**: The pre-trained 2D vision models can only process RGB images, and in the realm of robotics, there aren't widely-adopted pre-trained models for 3D point clouds. This necessitates the dependence on a separate non-pretrained network (i.e., another branch) to handle 3D point clouds. Interestingly, the approach you have suggested can be viewed as a variant of the two-branch framework. One branch (equivalent to SGR's semantic branch) extracts features from 2D images, while the second branch (akin to SGR's geometric branch and fusion network) not only processes 3D point clouds but also fuses the encoded 2D features with the features from 3D point clouds.
>
> > In the semantic branch, it seems that only RGB images from multiple views are used as inputs. Could the model benefit from processing the depth images from multiple views too?
>
> We do not use depth images in the semantic branch for two main reasons: (1) The image encoder in the semantic branch is exclusively pre-trained on RGB images and frozen in our approach. Consequently, integrating depth images directly into the semantic branch presents a challenge. (2) We utilize depth images in the geometric branch, where we convert them into point clouds using the camera's intrinsic and extrinsic parameters.
>
> > In the geometric branch, why is only the raw point cloud loaded, when the RGB colors of the points could be easily added for each point?
>
> Thank you for raising this point. Indeed, we did experiments with adding RGB colors for each point in the geometric branch, and the results are presented in the following table. Notably, this addition led to a reduction in performance.  We hypothesize that this decline in performance could be attributed to the lack of pre-training on colored point clouds, which might render the geometric branch susceptible to overfitting certain spurious RGB features.
>
> |                                 | open microwave | open door | water plants | toilet seat up | phone on base | put books | take out umbrella | open fridge | **Average** |
> |---------------------------------|:--------------:|:---------:|:------------:|:--------------:|:-------------:|:---------:|:------------------:|:-----------:|:---------:|
> | **SGR**                         |      52.6      |    80.2   |      40.2    |      80.1      |      81.1     |    84.8   |        94.7        |     34.8    |   68.6   |
> | **SGR w/ RGBs added for points**|      37.0      |    71.8   |      28.4    |      44.9      |      77.2     |    61.7   |        88.4        |      4.7    |   51.7   |

---

> > ### Author Response · Authors · 2023-08-11
> > **Response to Reviewer ebNV - Part 2**
> >
> > > The authors performed sim experiments on 8 RLBench tasks, but it seems that those tasks do not overlap the 18 RLBench tasks PerAct evaluated on. Could the authors pick a few tasks evaluated in PerAct and show at least some single-task quantitative results on them?
> >
> > Thank you for pointing out the differences in task choices between SGR and PerAct's evaluation on RLBench. Our selection of tasks is mainly based on their alignment with real-world situations. We use 8 tasks that closely mirror real-world scenarios, for example, our tasks incorporate objects like microwaves, plants, phones, books, and toilets that align more closely with the reality of domestic robot environments. Our aim is to leverage the powerful semantic understanding and spatial reasoning capabilities of SGR to facilitate the manipulation of a diverse and complex array of objects in real-world situations.
> >
> > However, acknowledging the significance of a direct comparison with PerAct, we did single-task experiments on 12 tasks that PerAct evaluated and the results are presented in the following table. Note that the results for PerAct were derived using their official codebase. The results indicate that SGR can still yield superior or at least comparable results. Particularly for tasks demanding complex semantic information, SGR exhibits a significant advantage.
> > |                          | open drawer | slide block | sweep to dustpan | meat off grill | turn tap | put in drawer | close jar | drag stick | screw bulb | put in safe | place wine | put in cupboard | **Average**   |
> > |--------------------------|:-----------:|:-----------:|:----------------:|:--------------:|:--------:|:-------------:|:---------:|:----------:|:----------:|:-----------:|:----------:|:---------------:|:-------------:|
> > | **SGR**                  | 79.9        | 97.7        | 84.9             | 94.0           | 87.4     | 57.1          | 36.3      | 96.5       | 9.5        | 30.6        | 54.9       | 11.6            | 61.7          |
> > | **PerAct**               | 87.3        | 96.4        | 32.9             | 97.9           | 5.5      | 4.4           | 23.2      | 75.3       | 14.7       | 7.9         | 39.7       | 7.9             | 41.1          |
> >
> >
> > > In the "generalizing to novel attributes" experiments presented in Figure 4, the PerAct baseline is missing. Is there a particular reason results on this baseline are not presented? It would be nice if the authors could run and present results on PerAct in this table.
> >
> > We genuinely appreciate your insightful feedback. Previously, in the generalization part of the paper, our main focus was to investigate the generalization ability of 2D representations versus 3D point-based representations. However, recognizing the importance of a thorough comparison, we have now incorporated the PerAct experiments as follows.
> >
> > The results indicate that PerAct shows comparable results on simple tasks, such as "reach target" and "reach shape" tasks, which require only a single keyframe action. However, it performs poorly on more complex tasks that require two keyframe actions, such as "slide block". Notably, SGR consistently showcases commendable generalization capabilities across these tasks, even those that are inherently complex.
> >
> > |          | reach target      |              | slide block      |              | reach shape      |              |
> > |----------|:-----------------:|:------------:|:----------------:|:------------:|:----------------:|:------------:|
> > |          | Seen              | Unseen       | Seen             | Unseen       | Seen             | Unseen       |
> > | **SGR**  | 94.0              | 36.8         | 97.7             | 43.0         | 99.0             | 27.7         |
> > | **PerAct**| 91.7            | 44.0         | 26.3             | 18.8         | 98.8             | 30.5         |

---

> > > ### Author Response · Authors · 2023-08-11
> > > **Response to Reviewer ebNV - Part 3**
> > >
> > > > It seems that the quantitative results of the real robot experiments are not very good. There indeed seems to be a gap between the performance of SGR and PerAct. However, out of the 8 tasks evaluated, SGR only outperforms PerAct in 4 of them.
> > >
> > > Thank you for noticing our quantitative results of the real robot experiments. While SGR and PerAct exhibit comparable performance across some tasks, SGR demonstrates superior performance in scenarios with visual distractors. For instance, in "Hitting Ball" and "Moving Cup to Goal" tasks (visual distractors can be viewed on our anonymous website), SGR achieves an impressive success rate of 70%, in stark contrast to PerAct's modest 30% achievement. For these two tasks, we observe that PerAct frequently manipulates the visual distractors rather than the objects specified during training. This observation leads us to hypothesize that while PerAct adeptly learns the concept of objects, it encounters challenges in differentiating between closely resembling objects.
> > >
> > > Furthermore, we want to underscore our meticulous approach to evaluating real robot experiments. Firstly, we meticulously maintain SGR and PerAct at the same initial conditions for each test episode, including scene configuration, robot pose, object locations, lighting, and other relevant factors. In addition to this, within each task, we initiate 10 episodes from diverse starting points. This ensures a broad and representative spectrum of tests, mitigating the potential biases from any specific initial positions. Through this diligent evaluation protocol, we secure the reliability and credibility of our reported results.
> > >
> > >
> > > > The paper lacks ablation experiments that study the extent to which parts of the method contribute to the overall performance. Some experiments that the authors could consider performing include (1) deleting one of the two branches in the network architecture while keeping other components of the framework;
> > >
> > > Thank you for highlighting the need for ablation experiments. Indeed, certain baseline models outlined in Section 4.1 can be interpreted as deleting one of the two branches in our SGR model. In particular, **PointNeXtLfS** and **CLIP** are designed to reflect scenarios where only the geometric or semantic branch is retained, respectively. Additionally, to ensure a fair comparison, we make sure all ablation models utilize the same input modalities (RGB-D images). In the case of PointNeXtLfS, we add RGB colors into the point cloud. For the CLIP model, we employ a separate 2D CNN to process depth images, and subsequently fuse the CLIP features and depth features together.
> > >
> > > > (2) switching out the pre-trained encoder with one that is trained from scratch.
> > >
> > > This aligns with the implementation of our **DenseFusion** in the paper. The results presented in Figure 4 demonstrate that replacing a pre-trained encoder with one that is trained from scratch will significantly reduce its generalization capabilities. Furthermore, to offer a more comprehensive perspective, we have supplemented with experimental results for **DenseFusion** under the single-task setting. The results also demonstrate the significant role of the pre-trained model within SGR.
> > >
> > > |                | open microwave | open door | water plants | toilet seat up | phone on base | put books | take out umbrella | open fridge |  **Average**  |
> > > |:--------------:|:--------------:|:---------:|:------------:|:--------------:|:-------------:|:---------:|:-----------------:|:-----------:|:-------:|
> > > | **SGR**        | 52.6           | 80.2      | 40.2         | 80.1           | 81.1          | 84.8      | 94.7              | 34.8        | 68.6    |
> > > | **DenseFusion**| 20.4           | 87.5      | 31.5         | 80.2           | 64.2          | 63.9      | 75.7              | 25.0        | 56.0    |
> > >
> > > We hope these address your concerns and showcase our thorough evaluation of the proposed methodology.

---

> > > > ### Author Response · Authors · 2023-08-16
> > > > **Do our responses answer your questions?**
> > > >
> > > > Dear Reviewer ebNV,
> > > >
> > > > We hope this message finds you well. As the author-reviewer discussion period is drawing to a close, we would like to inquire if our responses have addressed your concerns and questions to your satisfaction. If there are any remaining issues or further clarifications needed, we welcome the opportunity for additional discussions and are more than happy to provide further information. Thank you for your time and consideration.
> > > >
> > > >
> > > > Best Regards,
> > > >
> > > > The authors.

---

### Official Review · Reviewer_RMKE · 2023-07-20

**Confidence:** 4
**Originality:** Good
**Technical Quality:** Very Good
**Clarity Of Presentation:** Excellent
**Impact:** 3

**Recommendation:**

Weak Accept: I recommend accepting the paper, but will not argue for my recommendation if the majority of other reviewers have a different opinion.

**Review:**

Overall, the paper presented an interesting approach to integrate semantic and geometric representations, and claims that it stands as the only approach capable of generalizing to previously unseen semantic attributes. The paper is well written and the flow makes the paper easy to understand. However, there are a few points which should be addressed:

* The paper mentions that SGR achieves significantly higher performance when exposed to unseen colors or shapes. What happens in scenarios where a different shaped object requires a slightly different manipulation strategy? Does the method show signs of being able to adjust to different shapes in terms of the manipulation strategy?
* How is success measured? The authors mention that the real-robot experiments involve success out of 10 trials. However, the average success rate of SGR as compared to per act is only 11% higher, which is equivalent to one additional success. Because real-robot experiments are difficult to reproduce in general, and is prone to noise, such an increase isn't convincing.
* What is the runtime / computation resources needed by the method?

**Quality Of The Limitations Section:**

Limitations are addressed clearly

**Questions For Rebuttal:**

Overall, the paper was interesting to read. However, there are a few questions which I hope the authors could address:
* The paper mentions that SGR achieves significantly higher performance when exposed to unseen colors or shapes. What happens in scenarios where a different shaped object requires a slightly different manipulation strategy? Does the method show signs of being able to adjust to different shapes in terms of the manipulation strategy?
* How is success measured? The authors mention that the real-robot experiments involve success out of 10 trials. However, the average success rate of SGR as compared to per act is only 11% higher, which is equivalent to one additional success. Because real-robot experiments are difficult to reproduce in general, and is prone to noise, such an increase isn't convincing.
* What is the runtime / computation resources needed by the method?

**Robotics Focus:**

Sufficient demonstration on hardware

**Summary Of Paper:**

The paper presents Semantic-Geometric Representation (SGR), a perception module for robots that integrates rich semantic information from pre-trained 2D models with 3D spatial reasoning. The experiments demonstrate SGR's effectiveness in enabling robots to perform diverse simulated and real-world manipulation tasks, outperforming state-of-the-art methods in single-task and multi-task scenarios. The paper also highlights SGR's unique capability to generalize to novel semantic attributes.

**Summary Of Recommendation:**

The paper presents a compelling approach to integrating semantic and geometric representations, showing promising results in generalizing to previously unseen semantic attributes. The paper is well-written, and the flow enhances its readability. However, there are some aspects that need addressing, as mentioned earlier in the review section. Considering the interesting approach presented in the paper and its potential contributions to the field, I recommend a weak accept. However, addressing the above-mentioned points in a revised version would significantly strengthen the paper's overall quality.

---

> ### Author Response · Authors · 2023-08-11
> **Response to Reviewer RMKE**
>
> We thank Reviewer RMKE for their insightful and positive review.
>
> > The paper mentions that SGR achieves significantly higher performance when exposed to unseen colors or shapes. What happens in scenarios where a different shaped object requires a slightly different manipulation strategy? Does the method show signs of being able to adjust to different shapes in terms of the manipulation strategy?
>
> Thank you for your insightful question. In Appendix A, Figure 5(k) illustrates the "Reach Shape" task, wherein the robot is required to reach a point above a specified shape. This task primarily emphasizes visual perception, and currently, we are unable to directly observe the agent's complex interactions with various shapes. Nevertheless, we believe that our approach holds the potential to adapt to different shapes in terms of the manipulation strategy for the following reason. Our observations reveal that successful cases often involve the keyframe actions (target end-effector poses) predicted by our approach being positioned above the center of the object. Since objects of different shapes exhibit varying sizes and heights, this results in significant discrepancies in the predicted keyframe actions (target end-effector poses). As a result, when these actions are executed through a motion planner, they generate distinct robot trajectories based on the object's shape. In the future, we are eager to validate this potential further by employing more appropriate benchmarks and tasks, such as interacting with unseen objects within a certain category.
>
>
> > How is success measured? The authors mention that the real-robot experiments involve success out of 10 trials. However, the average success rate of SGR as compared to per act is only 11% higher, which is equivalent to one additional success. Because real-robot experiments are difficult to reproduce in general, and is prone to noise, such an increase isn't convincing.
>
> Thank you for raising this point on the measurement of success. Firstly, it is crucial to highlight that we meticulously maintain SGR and PerAct at the same initial conditions for each test episode, including scene configuration, robot pose, object locations, lighting, and other relevant factors. This meticulous approach ensures a direct and fair comparison between the two methods. Additionally, our experiments encompass 8 distinct tasks, and within each task, 10 episodes are initiated from diverse starting points. This ensures a broad and representative spectrum of tests, mitigating the potential biases from any specific initial positions. Moreover, to address the concerns about reproducibility, we have elaborated on the Real-Robot Setup in Appendix D. This detailed setup provides guidance and transparency, facilitating future attempts to replicate our results. We believe that even an 11% average improvement, while seemingly minor, is significant given the rigorous and consistent testing conditions we employed.
>
> > What is the runtime / computation resources needed by the method?
>
> Thank you for bringing up the vital topic of computational efficiency. SGR stands out significantly in this aspect, boasting impressive gains in both memory and time efficiency. In the single-task setting, SGR converges within 7 hours using a batch size of 32 and consumes a mere 17GB of GPU memory. This training can be efficiently executed on a single NVIDIA RTX 3090 GPU. In comparison, the voxel-based method, PerAct, can only accommodate a batch size of 2 on the same GPU and requires over 16 hours to converge. The noticeable disparity in efficiency stems mainly from SGR's design. Its semantic branch uses a frozen pre-trained 2D model, while the geometric branch adopts point-based models. These models exploit the sparsity of 3D point sets and inherently demand fewer computational resources than their voxel-based counterparts.

---

> > ### Comment · Reviewer_RMKE · 2023-08-15
> > **Thanks for the explanation**
> >
> > I've taken note of the authors' response, and they have addressed the inquiries I had put forward. I'll bear this in mind as I engage in discussions with the AC during the upcoming period.

---

### Decision · Program_Chairs · 2023-08-30

**Decision:**

Accept (Poster)

**Comment:**

The paper introduces a new method on learning semantic-geometric representation for robot manipulation.

After the rebuttal, reviewers still have diverge opinions on the paper. The remaining concerns of the reviewers are
- Experimental evaluation of the proposed method.
- Some claim of the paper is too strong.

Overall, the paper introduces a novel method which improves over previous state-of-the-art approaches.
Therefore, I recommend acceptance for the paper.
Due to the remaining concerns of the reviewers, I recommend the paper as a poster presentation.